# Large Language Models are Fixated by Red Herrings: Exploring Creative Problem Solving and Einstellung Effect using the Only Connect Wall Dataset

**Saeid Alavi Naeini**[1,3,4] **Raeid Saqur**[2,4] **Mozhgan Saeidi**[2,4,6] **John Giorgi**[2,4,5] **Babak Taati**[1,2,3,4]

[1]Kite Research Institute, Toronto Rehabilitation Institute, University Health Network
[2]Department of Computer Science, University of Toronto
[3]Institute of Biomedical Engineering, University of Toronto  [4]Vector Institute for AI
[5]Donnelly Centre for Cellular & Biomolecular Research, University of Toronto
[6]Department of Biomedical Data Science, Stanford University

{saeid.alavi, john.giorgi}@mail.utoronto.ca
raeidsaqur@cs.toronto.edu, mozhgans@stanford.edu, babak.taati@uhn.ca

## Abstract

The quest for human imitative AI has been an enduring topic in AI research since its inception. The technical evolution and emerging capabilities of the latest cohort of large language models (LLMs) have reinvigorated the subject beyond academia to the cultural zeitgeist. While recent NLP evaluation benchmark tasks test some aspects of human-imitative behavior (e.g., BIG-bench's 'human-like behavior' tasks), few, if not none, examine *creative problem solving* abilities. Creative problem solving in humans is a well-studied topic in cognitive neuroscience with standardized tests that predominantly use the ability to associate (heterogeneous) connections among clue words as a metric for creativity. Exposure to misleading stimuli — distractors dubbed *red herrings* — impede human performance in such tasks via the *fixation effect* and Einstellung paradigm. In cognitive neuroscience studies, such fixations are experimentally induced by pre-exposing participants to orthographically similar incorrect words to subsequent word-fragments or clues. The popular British quiz show Only Connect's *Connecting Wall* segment essentially mimics Mednick's Remote Associates Test (RAT) formulation with built-in, deliberate red herrings, which makes it an ideal proxy task to explore and study the fixation effect and Einstellung paradigm from cognitive neuroscience in LLMs. In this paper, we present the novel Only Connect Wall (OCW) dataset and report results from our evaluation of selected pre-trained language models and LLMs on creative problem solving tasks like grouping clue words by heterogeneous connections and identifying correct open knowledge domain connections in respective groups. We synthetically generate two additional datasets: `OCW-Randomized`, `OCW-WordNet` to further analyze our red-herrings hypothesis in language models. The code and link to the dataset are available at `https://github.com/TaatiTeam/OCW`.

## 1  Introduction

The remarkable capabilities of state-of-the-art large language models (LLMs) [91], across a variety of domains and downstream tasks [78, 10], have spurred their comparisons with artificial general intelligence (AGI) [5, 14] and human-imitative AI [31] systems. The extraordinary leap in capabilities of these LLMs over a short span — from the advent of transformer-based [69] pre-trained, context-aware language models (PLMs) [52, 17, 40, 36, 53] circa 2018 to 2020, to the current and latest

37th Conference on Neural Information Processing Systems (NeurIPS 2023) Track on Datasets and Benchmarks.

**Wall A: Season 11, Episode 23**

| Gala | Twelfth | Bonfire | Hen | ___ night |
|---|---|---|---|---|
| Orlov | Churchill | Digby | Tony | Advert Animals |
| Burns | Marx | Clarke | Bender | Cigar Smokers |
| Canal Street | Castro | Chelsea | Darlinghurst | Gay Villages |

**Wall B: Season 12, Episode 27**

| Jazz | Gala | Honeygold | Jonathan | Apples |
|---|---|---|---|---|
| Healing | Join | Greg | Show of | Can Precede Hands |
| Pippin | Merry | Gaffer | Sam | Hobbits |
| Twill | Duct | Ticker | Cassette | Types of Tape |

**Wall C: Season 15, Episode 10**

| Cameo | Fuji | Bramley | Jazz | Apples |
|---|---|---|---|---|
| Amy | Lady Bird | Dakota | Dwayne | Johnsons |
| Thunder | Magic | Heat | Celtics | US Basketball Teams |
| Gala | Costume | Goggles | Pool | Swimming ___ |

**Wall D: Season 10, Episode 2**

| Shrewsbury | Wellington | Ludlow | Madeley | Shropshire Towns |
|---|---|---|---|---|
| Bath | Boarding | Doge | Cathode | Begin with Animals |
| Chelsea | Gum | Snow | Cowboy | Boots |
| Bolt | Bond | Churchill | Coward | English Playwrights |

Figure 1: Examples of *Only Connect* walls with ground-truth groupings (rows) and connections (last column). *Red herrings* include orthographically identical words, e.g., **Gala**, **Churchill** and **Chelsea** in different connected groups — **Gala**: *Gala night, Apples*, *Swimming gala*, **Churchill**: *Advert Animals, English Playwrights* and **Chelsea**: *Gay Villages, Boots* — across walls. In Wall **A** (top left), the clues **Churchill**, **Marx**, **Castro** provide misleading stimuli inducing plausible fixation on historical figures within the wall.

cohort of increasingly larger (billions of parameters) LMs [59, 77, 57, 89, 16, 67, 18] spearheaded by the OpenAI's GPT series [13], notably ChatGPT [49] and GPT-4 [48] — justifiably warrants such comparisons. Several natural language processing (NLP) benchmarks have been proposed to standardize the evaluation of these LLMs, including MMLU [27], BIG-bench [66], HELM [38], and Global-Bench [65]. The tasks inventory under these benchmarks are open (type of tasks) and dynamic (rolling additions). While a subset of these tasks aims to test for human imitative intelligence (e.g., nineteen tasks listed under the *human-like behaviour* category in BIG-bench), none tests for *creative problem solving* abilities [44] — a hallmark of human-like intelligence [31].

Creative problem-solving by humans is a well-studied topic in cognitive neuroscience and human behavioural sciences literature. These studies and methods use (word) associative fluency to model and test creativity objectively [44, 9]. Empirical research in this context commonly employs single or continuous word association tests that are variants of Mednick's seminal Remote Associates Test (RAT) [45]. Such tests entail finding connections or links among a presented group of words using associations that can be heterogeneous (e.g., synonymy, semantic, compounding) [86, 43]. To exemplify, consider the cue words: *{Tennis, Same, Head}*. A correct connection in this triplet is: *Match*, which connects by semantic link (*tennis match*), synonymy (*same match*), and compounding (*match head*). Further, the word connections can also vary in degrees of figurativeness (e.g., *Star-Actress* vs. *Star-Planet*) and abstractness (e.g., *Humor-Sense* vs. *Apple-Tree*). In humans, such creative problem-solving abilities are impeded by exposure to wrong answers [61, 62, 85] — a finding referred to as the *fixation effect* [34, 82]. A closely related similar concept is the *Einstellung effect* [42], which postulates the negative effect of previous experience when solving new problems.

Studies examining the fixation effect induce fixations by presenting clue words intended as wrong answers (misleading stimuli) [61] dubbed "red herrings" or, by pre-exposing participants to red herrings before attempting creative problem-solving tasks like the RAT [45]. A slew of works in negative transfer learning in human cognition attempt to explain the RAT fixation phenomenon that involves pre-exposure to red herrings by the negative effects of prior learning on indirect or implicit measures of memory [63]. This negative transfer effect was demonstrated and studied using orthographically similar words to subsequent test word fragments as red herrings [63]. Intuitively, the red herrings lead participants away from the memory retrieval (or down incorrect neurological pathways by Hebbian terminology) required for correct responses and fixate on wrong connections [60]. Fixation in creative problem-solving can be increased by making red herrings more retrievable. Thus, creative problem-solving can be thought of as a type of indirect memory measure whose retrieval is degraded by red herrings due to the negative transfer effect. The *red herring retrieval hypothesis* states that factors that make red herrings more retrievable should reduce creative problem-solving performance,

as measured with RAT problems. Two such factors are repetition and context. A following corollary states that the memory strengths of red herrings determine the magnitude of a fixation effect [8].

In this work, we study the juxtaposition of these theories from human cognitive neuroscience (*fixations, negative transfer learning, red herring memory retrieval hypothesis*) from the context of LLMs and natural language processing. While negative transfer learning has been observed and studied in AI research [76, 22], the context of these studies is limited to strict machine learning sub-domains like statistical distribution measures and computer vision. There has not been any work that systematically examines these specific concepts' relation in AI research. Our major contributions are as follow:

**1. Only Connect Wall (OCW) Dataset and creative problem solving tasks.** We introduce a novel dataset for evaluating *creative problem solving* tasks by curating the problems and human performance results from the popular British quiz show Only Connect [81, 3]. Specifically, the *Connecting Walls* segment of the show, where the tasks entail grouping sixteen (16) jumbled up clue words into four (4) connected groups, and naming the correct connections (Figure 1). The presented words have heterogeneous connections with open-domain knowledge retrieval, e.g., history, places, famous people, tools, and cultural references. These 'walls' contain red herrings or misleading stimuli by design, which makes this dataset an analogical proxy for RAT tests in evaluating LLMs for creative problem-solving. Section §2 provides a detailed description of the dataset.

**2. Experiments, results, and key findings of baseline LLMs evaluation.** We evaluate a suite of NLP models from static embeddings to PLMs to LLMs and demonstrate that none can solve the tasks of the OCW dataset. Our findings show that SOTA LLMs (e.g. GPT-4 [48]) perform significantly worse than the expert human baseline, and somewhat surprisingly, that increasing the number of in-context examples in few-shot in-context-learning is ineffective. Sections §3 and §4 provide details.

## 2   Only Connect Walls Dataset

Here we focus on the *Connecting Walls* segment (usually the third round) of the quiz-show. Each wall contains sixteen jumbled-up word clues that must be sorted into four groups, each with four connected words. Once the groups are formed, contestants must also identify the right connection or relationship among the items in each group. While there is only one correct solution to each wall, the puzzles are designed to include several red herring clues that can fit into another category and red herring categories fitting multiple clues. Figure 1 shows solved sample walls from the show highlighting a couple of typical red herrings.

### 2.1   Dataset Collection and Structure

The OCW dataset contains 618 connecting wall puzzles and solutions in total from 15 seasons of the show. Each show episode has two walls. The total number of walls per season varies based on the (varying) number of aired season episodes. The walls were scraped from fan websites[1], and human performance results (for grouping and connection tasks) were manually curated by watching all the episodes. Figure 2 depicts the high-level structure of the dataset in JSON format with self-explanatory object keys and comments.

### 2.2   Tasks and Evaluation Metrics

The two dataset tasks: **Task 1 (Grouping)**, and **Task 2 (Connections)** are identical to the quiz-show's human participant tasks. We evaluate Task 1 (Groupings) via six metrics: number of solved walls, number of correct groups (max. four per wall), Adjusted Mutual Information (AMI) [71], Adjusted Rand Index (ARI) [28], Fowlkes Mallows Score (FMS) [21], and Wasserstein Distance (WD) [54], normalized to $(0, 1)$ range, between predicted and ground-truth labels [88, 70].

We similarly evaluate Task 2 (Connections) with three metrics: exact string matching, ROUGE-1 F1 [39], and BERTScore F1 [90]. Exact match is the most strict, assigning a score of 1 when the predicted connection is identical to the ground-truth and 0 otherwise. ROUGE-1 F1 relaxes this criterion; it is large when there is a high proportion of ground-truth tokens in the model's predicted

---

[1]The primary source was the Only Connect fan website: `https://ocdb.cc` [6].

| Predicted Connection | Ground-truth Connection | Exact Match | ROUGE-1 F1 | BERTScore F1 |
|---|---|---|---|---|
| Types of numbers | Types of numbers | 1.00 | 1.00 | 1.00 |
| Slang terms for money | Slang for money | 0.00 | 0.86 | 0.79 |
| Types of trees | Trees | 0.00 | 0.50 | 0.63 |
| Bridges in London | Thames bridges | 0.00 | 0.40 | 0.31 |
| Medieval occupations | Chaucer characters | 0.00 | 0.00 | 0.15 |

Table 1: Examples of predicted and ground-truth connections and their performance according to the chosen metrics. Exact match is 0 for anything but identical strings. Empirically, we observe that a ROUGE-1/BERTScore F1 of $\geq 0.5$ indicates that a predicted connection is likely *correct*.

connection *and* a low proportion of non-ground truth tokens. BERTScore F1 is similar but further relaxes this criterion, assigning a non-zero score for *semantically* similar (but non-identical) predicted tokens. Together these three metrics provide a more holistic view of model performance on Task 2 than any one metric alone. Empirically, we find that a ROUGE-1 or BERTScore F1 of $\geq 0.5$ indicates that a predicted connection would likely be considered *correct* (Table 1). Note that BERTScore has many parameters affecting the final score; a hashcode is produced and reported for reproducibility.

Each of the evaluation metrics for Task 1 of Task 2 could be calculated per wall, per episode, per season, or for the entire test set. We present results on the entire test set in this paper (§4). We split the dataset into a train set (62 walls), validation set (62 walls), and test set (494 walls). The primary goal of our dataset is to evaluate the zero- and few-shot creative problem-solving abilities of LLMs; as such, we elect to set the size of the test set to be much greater than train or validation sets.

## 3    Experiments: Language Model Evaluations

This section describes methods and models used to provide baseline results for the dataset. For Task 1 (Grouping), we use clustering techniques on word-embeddings from classical and pre-trained language models (PLMs) (§3.1), and few-shot in-context learning (ICL) with LLMs (§3.2). For Task 2 (Connections), we only provide baseline results using few-shot ICL with LLMs (§3.2).

```
{// Contains a mapping of number of walls per season
"season_to_walls_map": {"1": 30, "2": 16, ...},
// Contains the actual dataset, one object per wall
"dataset": [{
  // Each wall has a unique identifier, season and episode number
  "wall_id": "3ca8", "season": 1, "episode": 8,
  // The list of 16 words or "clues" associated with the wall
  "words": ["Holmes", "Indiana Jones", "Bannister", ...],
  // Ground-truth connections for each of the four groups of this wall
  "gt_connections": ["Parts of a staircase", "___ cake", ...],
  // The four ground-truth groups for this wall
  "groups": {
    // Each group is an object with...
    "group_1": {
      // ...a unique ID
      "group_id": "3ca8_01",
      // ...ground-truth words
      "gt_words": ["Newel", "Bannister", "Tread", "Riser"],
      // ...and ground-truth connection
      "gt_connection": "Parts of a staircase",
      // Human performance is recorded as solved (1) or unsolved (0)
      "human_performance": {"grouping": 1, "connection": 1}
    }, ...
  }
  // Overall human performance for each group within the wall
  "overall_human_performance": {
    "grouping": [1, 1, 1, 1],
    "connections": [1, 1, 1, 0],
  }
} ...]}
```

Figure 2: JSON Structure of the OCW dataset. One truncated example is shown.

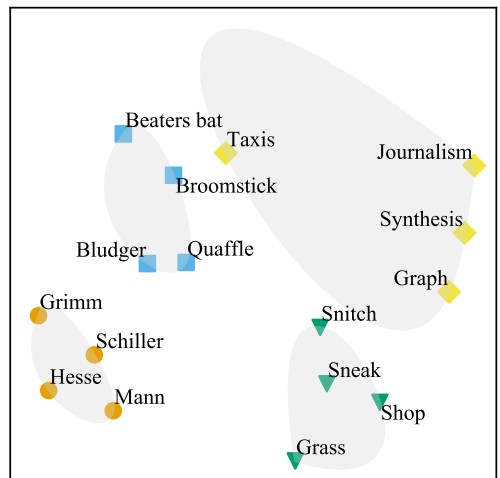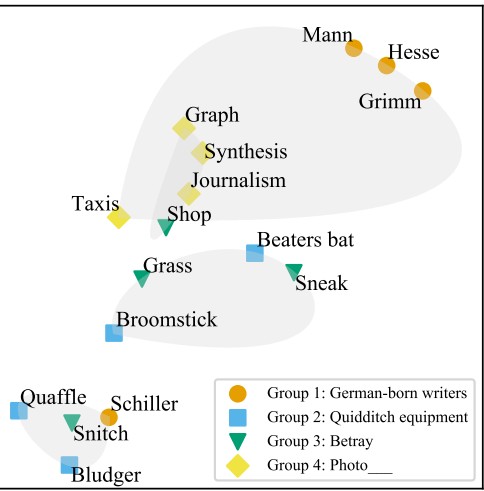

Figure 3: Solved wall (`wall_id="8cde"`) for Task 1 (Grouping) using best performing model ($E5_{BASE}$) with both static and contextual embeddings. **Left**: solved wall using static embeddings. **Right**: unsolved wall using contextual embeddings. 2D projection of embeddings using t-SNE is shown. Colors and shapes correspond to true clusters, and grey convex regions correspond to predicted clusters. The legend shows the ground truth connection for each group.

## 3.1 Task 1: Grouping using Word Embeddings

For the *grouping task* evaluation (§2.2), we use clustering algorithms on word-embeddings of the sixteen clue words in each wall, to group them into four predicted groups that are subsequently evaluated against the four ground-truth groups for each wall. A vanilla $k$-means (with $k = 4$) clustering algorithm [25] does not guarantee each predicted group to have four words, thus we use variants like constrained clustering.

**Clustering**    Semi-supervised constrained clustering [72, 7] is used when the user has pre-existing knowledge about the desired partition (in our case, 4 groups). Here, we adopt a *minimum cost flow network* clustering approach [12] with a cluster size of four for grouping. Our preliminary analysis showed that clustering results exhibited slight variations across runs. This slight discrepancy could be attributed to the initializations of cluster centroids. To address this issue and ensure reliable results, we report the mean and variance of results (Table 3)across sixteen (16) runs, each with a unique seed and randomized order of sixteen-word clues. We tested two additional clustering approaches motivated by [47, 19]: (1) We constructed a self-similarity matrix containing pair-wise similar information about the words prior to applying constrained clustering; (2) We performed dimensionality reduction using Principal Component Analysis (PCA) [58] and t-distributed stochastic neighbor embedding (t-SNE) [68] before applying constrained clustering. Neither approach improved performance over raw embeddings' clusters, and, for brevity, results are not included.

**Static word embedding**    We used two well-known classic word embedding models, GloVe [51] and FastText [23], both of which are accessed through the Flair library (Table 2). We used two FastText models, one pre-trained on the Common Crawl corpus and another on Wikipedia. Approximately 10% of the total clues encountered in the dataset were out-of-vocabulary (OOV). A significant portion (~80%) of the OOV cases were addressed by mean pooling for clues comprised of multiple words to obtain one unified embedding. For the remaining OOV instances, we combined the static embeddings with BytePair encoded[26] sub-words.

**PLMs**    We explored general-purpose PLMs (BERT [17], RoBERTa [40], DistilBERT [56], ELMo [52]) as well as Sentence Transformers (MPNet [64], E5 [75]; see Table 2). We evaluated performance with and without contextual embeddings.[2] Depending on the context, some clues in the dataset may appear across different walls with different meanings. As an example, the word

---

[2]Static embeddings are obtained from the PLMs by passing clues through the model *independently*.

| Model | | # Parameters | Version | Accessed via |
|---|---|---|---|---|
| | | | *Word Embeddings* | |
| BPEmb [26] | En | – | `en` | Flair [4] |
| GloVe [51] | 6B | – | `glove` | Flair |
| FastText [23] | Crawl | – | `crawl` | Flair |
| | News | – | `news` | Flair |
| | | | *Pre-trained Language Models (PLMs)* | |
| ELMo$_{LARGE}$ [52] | | 94M | `large` | Flair [4] |
| DistilBERT$_{BASE}$ [56] | uncased | 66M | `distilbert-base-uncased` | HuggingFace [83] |
| BERT$_{BASE}$ [17] | uncased | 110M | `bert-base-uncased` | HuggingFace |
| BERT$_{LARGE}$ | uncased | 340M | `bert-large-uncased` | HuggingFace |
| RoBERTa$_{LARGE}$ [40] | | 355M | `roberta-large` | HuggingFace |
| | | | *Sentence Transformers* | |
| all-mpnet$_{BASE}$ [64] | V2 | 110M | `sentence-transformers/all-mpnet-base-v2` | HuggingFace |
| E5$_{BASE}$ [75] | V2 | 110M | `intfloat/e5-base-v2` | HuggingFace |
| E5$_{LARGE}$ | V2 | 340M | `intfloat/e5-large-v2` | HuggingFace |
| | | | *Large Language Models (LLM)* | |
| GPT-3.5-turbo | | – | `gpt-3.5-turbo-0301` | OpenAI API |
| GPT-4 | | – | `gpt-4-0314` | OpenAI API |

Table 2: Details about the baselines and models used in our experiments.

"Gala" was found in three distinct walls, each associated with a different meaning: *apples*, *swimming*
___, and ___ *night* (Figure 1). The contextual embeddings were aimed to capture contextual semantic
similarity among the clues (if any). They were generated by joining the 16 clues in the wall as a
pseudo-sentence. We randomly shuffle the word order across sixteen different runs for each wall to
account for the positional ordering. We note that such faux sentences (for inducing context) are not
valid English syntactic sentences. We used mean pooling to generate embeddings for clues comprised
of multiple words to capture the collective meaning of the entire clue.

## 3.2 Task 2: Connections using Few-shot In-context Learning (ICL) with LLMs

Few-shot ICL with LLMs has emerged as a performant and broadly applicable paradigm in NLP [13].
To evaluate the performance of this approach on our proposed dataset, we designed a few-shot prompt
for GPT-3.5-turbo and GPT-4 [48], which are amongst the strongest performing LLMs currently
available.[3] For Task 1 (Grouping, §2.2), the prompt consists of some natural language instructions,
several examples of solved walls from the training set, and the current example's 16 clues, randomly
sorted. For Task 2 (Connections), in place of the 16 clues, the prompt contains a solved wall *without*
the connections (Figure 4).

We developed our prompts on the validation set and reported the final performance on the test set.
In-context examples are randomly selected from the train set; the same examples are used across
all test inputs. We experiment with 0, 1, 3, 5, and 10 in-context examples. When necessary, we
apply simple post-processing to the LLMs output. For example, in both Task 1 and Task 2, we
take a maximum of 4 predictions for the groups and connections, respectively, and pad up to 4 with
the empty string in cases where the model outputs fewer than 4.[4] To make results as reproducible
as possible, we set the `temperature=0` and used the 03/01/2023 GPT-3.5-turbo snapshot and the
03/14/2023 GPT-4 snapshot. The max output length is set to 144 tokens. All other hyperparameters of
the OpenAI API are left at their defaults [2]. Prompts were designed as per the Guidance library [1].

## 4 Results and Discussions

### 4.1 Task 1: Grouping Results

**Embedding Clustering Techniques** In Table 3 we report the performance of several static embed-
ding baselines on Task 1 (Grouping). E5$_{BASE}$ was the most performant model and, on average, solved

---

[3]In preliminary experiments, we found that open-source LLMs like LLaMA [67] perform poorly and
typically do not follow the task instructions.

[4]Please see our codebase for all post-processing steps: `https://github.com/TaatiTeam/OCW`

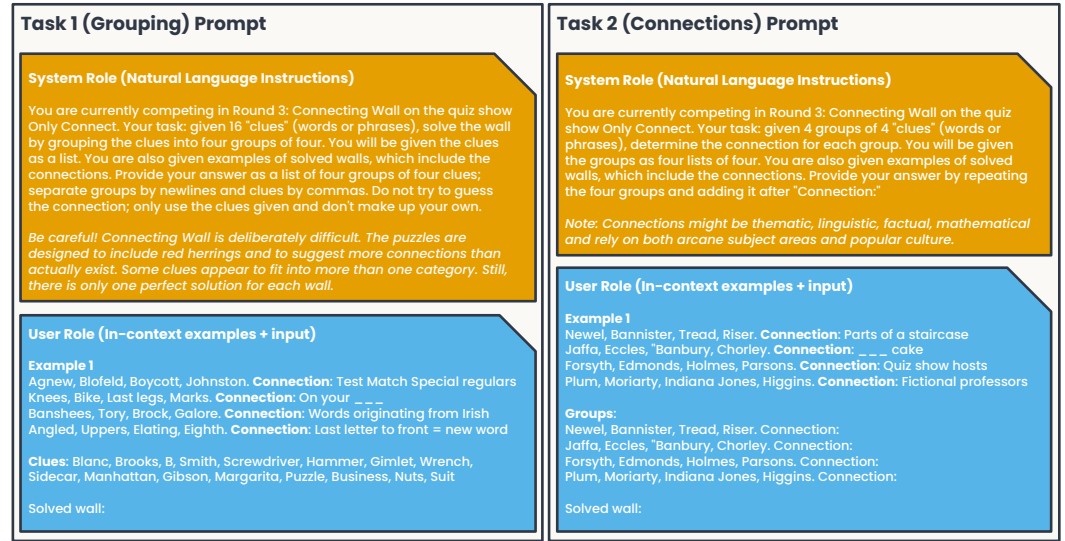

Figure 4: Example prompts for Task 1 (Grouping) and Task 2 (Connections) used with GPT-3.5-turbo and GPT-4. The system's role includes natural language instructions. The user role includes $n$ in-context examples and the current examples 16 clues (Task 1) or the solved wall without connections (Task 2). For Task 1, the model is instructed to output the solved wall as four lines of four clues separated by commas. For Task 2, the model is instructed to copy the solved wall and fill in the connections. *Emphasis* and **bold text** are for visualization purposes only.

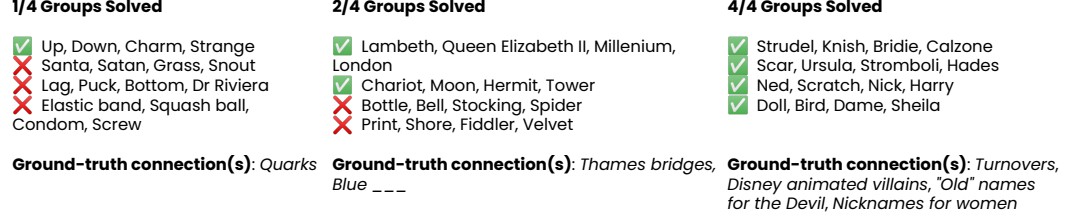

Figure 5: Examples of partially and fully solved walls predicted by GPT-4.

1 wall and correctly clustered 89 groups. Contextual embeddings had the lowest overall performance amongst all methods (Table 6 in Appendix §A). One explanation is that by concatenating all the clues in each wall, the resulting input may not adhere to the sentence structure that PLMs are accustomed to during training. This may disrupt the natural flow of information and, in turn, lead to less meaningful contextual embeddings. Moreover, the context may change abruptly when combining clues from different parts of the wall. This can introduce ambiguity and contextual shifts that the model may struggle to interpret accurately. Another possible explanation is the effect of positional encoding in the underlying models. Unlike other main components of PLMs, positional encoding is variant to sequence order [33]. Even though the addition of positional to word embeddings helps with learning the contextual representation of words at different positions, intrinsic similarities may be more important than contextual usage. The embedding dependence on neighboring clues may have hindered the clustering process by introducing noise and capturing irrelevant information that is specific to a particular context. For instance, in Figure 3, the contextual embedding model erroneously associated the clue "*Shop*" with the connection "*Photo ___*", resulting in the formation of the word "*Photoshop*"; however, this association is incorrect as it is an example of a red herring in the wall. In contrast, the static embedding model correctly mapped "*Shop*" to its British slang meaning connection "*Betray*". Please refer to Appendix §B for more examples.

**Few-shot ICL with LLMs** Performance of GPT-4 far surpassed the static (Table 3) and contextual embedding baselines (Table 6), particularly in terms of the number of solved walls and correct groups (>2X the next most performant model, E5), but was still far below human performance (§4; see §5 for example predictions). Examining the predictions of the best-performing model (GPT-4,

| | WD ↓ | FMS ↑ | ARI ↑ | AMI ↑ | # Solved Walls | # Correct Groups |
|---|---|---|---|---|---|---|
| *Classic Word Embeddings* | | | | | | |
| GloVe | $84.9 \pm .4$ | $31.5 \pm .3$ | $14.4 \pm .3$ | $17.6 \pm .4$ | $0 \pm 0$ | $68 \pm 4$ |
| FastText (Crawl) | $84.2 \pm .5$ | $32.1 \pm .3$ | $15.2 \pm .3$ | $18.4 \pm .4$ | $0 \pm 0$ | $80 \pm 4$ |
| FastText (News) | $85.5 \pm .5$ | $30.4 \pm .2$ | $13.0 \pm .2$ | $15.8 \pm .3$ | $0 \pm 0$ | $62 \pm 3$ |
| *Pre-trained Language Models (PLMs)* | | | | | | |
| $ELMo_{LARGE}$ | $86.3 \pm .6$ | $29.5 \pm .3$ | $11.8 \pm .4$ | $14.5 \pm .4$ | $0 \pm 0$ | $55 \pm 4$ |
| $DistilBERT_{BASE}$ | $86.7 \pm .6$ | $29.1 \pm .2$ | $11.3 \pm .3$ | $14.0 \pm .3$ | $0 \pm 0$ | $49 \pm 4$ |
| $BERT_{LARGE}$ | $88.3 \pm .5$ | $26.5 \pm .2$ | $8.2 \pm .3$ | $10.3 \pm .3$ | $0 \pm 0$ | $33 \pm 2$ |
| $BERT_{BASE}$ | $89.5 \pm .4$ | $25.1 \pm .2$ | $6.4 \pm .3$ | $8.1 \pm .4$ | $0 \pm 0$ | $22 \pm 2$ |
| $RoBERTa_{LARGE}$ | $88.4 \pm .4$ | $26.7 \pm .2$ | $8.4 \pm .3$ | $9.4 \pm .4$ | $0 \pm 0$ | $29 \pm 3$ |
| *Sentence Transformers* | | | | | | |
| $all\text{-}mpnet_{BASE}$ | $86.3 \pm .4$ | $29.4 \pm .3$ | $11.7 \pm .4$ | $14.3 \pm .5$ | $0 \pm 0$ | $50 \pm 4$ |
| $E5_{LARGE}$ | $84.4 \pm .7$ | $32.3 \pm .4$ | $15.4 \pm .5$ | $18.5 \pm .6$ | $0 \pm 0$ | $76 \pm 5$ |
| $E5_{BASE}$ | $\mathbf{83.8 \pm .6}$ | $\mathbf{33.1 \pm .3}$ | $\mathbf{16.3 \pm .4}$ | $\mathbf{19.5 \pm .4}$ | $\mathbf{1 \pm 0}$ | $\mathbf{89 \pm 6}$ |
| Human Performance | – | – | – | – | 285 / 494 | 1405 / 1976 |

Table 3: Results of selected models on Task 1 (Grouping) using static embeddings. WD: Wasserstein Distance. FMS: Fowlkes Mallows Score. ARI: Adjusted Rand Index. NMI: Normalized Mutual Information. Mean ± standard deviation over 16 random seeds is shown. **Bold**: best scores.

| | # In-context Examples | WD ↓ | FMS ↑ | ARI ↑ | AMI ↑ | # Solved Walls | # Correct Groups |
|---|---|---|---|---|---|---|---|
| GPT-3.5-turbo | 0-shot | 82.5 | 34.0 | 18.4 | 21.6 | 0 | 114 |
| | 1-shot | 82.3 | 34.4 | 18.2 | 21.2 | 0 | 123 |
| | 3-shot | 80.9 | 36.8 | 21.3 | 24.7 | 0 | 140 |
| | 5-shot | 80.6 | 37.3 | 22.0 | 25.4 | 2 | 149 |
| | 10-shot | 81.2 | 36.1 | 20.4 | 24.0 | 2 | 137 |
| GPT-4 | 0-shot | 75.8 | 41.5 | 27.2 | 30.7 | 6 | 239 |
| | 1-shot | 73.4 | 43.7 | 29.7 | 33.5 | 4 | 262 |
| | 3-shot | 73.7 | **43.9** | **29.9** | **33.6** | 5 | **272** |
| | 5-shot | **72.9** | 43.4 | 29.1 | 32.8 | **7** | 269 |
| | 10-shot | 73.6 | 42.8 | 28.5 | 32.3 | 3 | 249 |
| Human Performance | | – | – | – | – | 285 / 494 | 1405 / 1976 |

Table 4: Results on Task 1 (Grouping) using Large Language Models. WD: Wasserstein Distance. FMS: Fowlkes Mallows Score. ARI: Adjusted Rand Index. NMI: Normalized Mutual Information. **Bold**: best scores.

3-shot), we found common sources of error to include misformatted outputs (4.4% of all predicted groups) and hallucinated clues (6.6%).

Surprisingly, more in-context examples (from 1 to 10 shot) did not improve performance. One possible explanation for this observation is that, due to the huge variety of possible connection types, the in-context examples' primary benefit is demonstrating the expected output format – as opposed to demonstrating how to perform the task – which likely requires only a single example. This is related to the concepts of *task learning* versus *task recognition*, which are thought to be the two distinct mechanisms through which ICL leverages demonstrations [50, 32]. Many clues require open-domain, arcane, cultural and intimate knowledge of niche subject areas (e.g., "*Professional snooker players*", "*Female Radio 1 DJs*") that, without prior memorization, are unlikely to help. The presence of orthographically similar clue words in the in-context examples could themselves act as red herrings and plausibly induce negative transfer learning. An interesting future direction would be the evaluation of retrieval augmented models [24, 37, 11, 29], which may be capable of solving groups about highly specific subject areas.

## 4.2 Task 2: Connections Results

In Figure 6, we present the results for Task 2 (Connections). In general, GPT-4 outperforms GPT-3.5-turbo, especially in the 0-shot regime. Performance for GPT-4 improves monotonically with an increasing number of in-context examples, although improvements are sometimes small (e.g.,

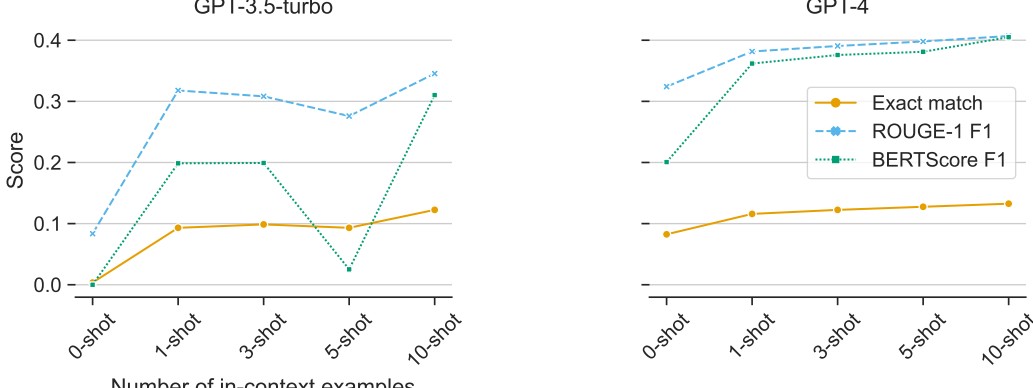

Figure 6: Results for Task 2 (Connections) with GPT-3.5-turbo and GPT-4. For reference, human performance is approximately 80% (fraction of correctly answered connections). We report $\max(\text{BERTScore}, 0)$ in the case of GPT-3.5-turbo for readability.

$< 0.01$). As expected, the exact match score for both models is low ($< 15\%$). This is explained by the fact that even insignificant differences between the model's predictions and the ground truth will result in a score of 0 (e.g., "Made *of* rubber" vs. "Made *from* rubber"). For this reason, we also report ROUGE-1 and BERTScore F1 scores (§2.2). Although not a perfect comparison, we can contextualize these results with human performance, which we recorded as the fraction of correctly guessed connections: ∼80% on the test set. The quiz show Only Connect allows for some small deviations in guessed connections that will be accepted as correct, making the comparison to ROUGE and BERTScore more suitable than to exact match. Our results suggest that at 41-45% F1, the best performance achieved with few-shot ICL (GPT-4, 10-shot) is far below human performance. Lastly, we note that a common source of model error was the inclusion of clues in the predicted connection (occurring in 8.2% of all predicted connections for the best performing model), e.g., "Fireplace tools (Spade, Brush, Poker, Tongs)", even though (1) the model was not instructed to do so, and (2) the in-context examples were not formatted like this.

More complicated post-processing or prompting strategies (e.g., "Chain of Thought" [79], "Tree of Thoughts" [87]) could mitigate these issues and improve performance. However, applying these more complicated prompting strategies to the OCW dataset is non-trivial, as they require breaking down the problem into intermediate steps, and the number or nature these intermediate reasoning steps should take is unclear. We leave their application to the OCW dataset for future work.

### 4.3 Effects of Red-Herrings: Additional Datasets, Experiments and Analyses

To analyze our *red-herring hypothesis* on language models, we designed and performed additional ablative experiments. The original `OCW` dataset contains red-herrings as distractors *by design*. We generate two additional datasets from `OCW` to decrease the presence of red-herrings: `OCW-Randomized` and `OCW-WordNet`. The goals, construction and other details are presented in Appendix §C.1.

In `OCW-Randomized`, we diluted the presence of red herrings by randomly swapping groups among the walls in the test set – thus negating the inherent deliberate distractor groups in each wall. We further simplify the grouping task in `OCW-WordNet` by removing red herrings altogether. This is achieved by using subordinate-superlative (or hyponym-hypernym) word hierarchy and synonyms in the English lexical database WordNet [46, 20]. Thus the results in Table 5 present results on datasets with a decreasing proportion of red herrings from left to right, and by our hypothesis, increasing task simplicity for LLMs. The results are aligned with our expectations, with GPT-3.5-turbo and GPT-4 performance increasing significantly with the reduction of red herrings from the test set.

## 5   Related Work

Various datasets and tasks have been proposed for evaluating language models against human-like linguistic capabilities. Earlier examples of such tasks include *word sense disambiguation* (WSD) [55],

|  |  | OCW | | OCW-Randomized | | OCW-WordNet | |
| --- | --- | --- | --- | --- | --- | --- | --- |
|  |  | # Solved Walls | # Correct Groups | # Solved Walls | # Correct Groups | # Solved Walls | # Correct Groups |
| GPT-3.5-turbo | 0-shot | 0 | 114 | 5 | 274 | 337 | 1522 |
|  | 1-shot | 0 | 123 | 12 | 315 | 320 | 1400 |
|  | 3-shot | 0 | 140 | 10 | 306 | 415 | 1748 |
|  | 5-shot | **2** | **149** | 16 | **337** | 415 | 1759 |
|  | 10-shot | 2 | 137 | **17** | 333 | **428** | **1800** |
| GPT-4 | 0-shot | 6 | 239 | 59 | 595 | **471** | **1926** |
|  | 1-shot | 4 | 262 | 57 | 644 | 304 | 1581 |
|  | 3-shot | 5 | **272** | 62 | 649 | 279 | 1537 |
|  | 5-shot | **7** | 269 | **68** | **655** | 298 | 1584 |
|  | 10-shot | 3 | 249 | 55 | 614 | 378 | 1742 |
| Human Performance |  | 285 / 494 | 1405 / 1976 | – | – | – | – |

Table 5: Coalesced results of LLMs performance on Task 1 (grouping) using two additional test datasets OCW-Randomized and OCW-WordNet with decreasing presence of red-herrings from *left* to *right* in the walls, juxtaposed against the original OCW test set (left-most column). Only the main metrics are shown (details and full results in Appendix §C). **Bold**: best scores.

Winograd schema challenge [35] and *word sense induction* (WSI) [80]. WSD aims to determine a word's correct meaning or sense within a specific context. WSI focuses on automatically clustering words into different senses or semantic categories based on their contextual usage patterns. Benchmarks like GLUE [74] and SuperGLUE [73] are aimed at aggregating and standardizing these classical NLP tasks to evaluate language models. The PLMs (e.g., BERT variants) and the first generation of LLMs, mostly solved or attained human-level performance on these tasks by 2020s [41].

In order to evaluate the human-imitative capabilities of modern LLMs, more challenging tasks have been proposed in recent benchmarks like BIG-bench [66] and HumanEval [15]. **BIG-bench** aims to address the limitations of existing benchmarks by providing a more comprehensive, open, and dynamic (tasks added on a rolling basis) evaluation benchmark. It covers a wide range of tasks, including a suite of tasks targeted specifically for *human-like behavior*. **HumanEval** is an evaluation set to measure the functional correctness of code synthesis from docstrings [15]. This benchmark includes 164 original programming problems that assess language comprehension, algorithms, and simple mathematics comparable to simple software interview questions. While these recent benchmarks include a wide net of complex tasks, evaluating a broad range of LLM capabilities, our work here is orthogonal to these since none of them aims to specifically measure creative problem-solving or creativity and their impediments in LLMs.

## 6 Limitations & Future Work

As with any machine learning dataset, especially one designed to evaluate the performance of LLMs, the OCW dataset has several limitations. First, we noticed that the performance of contextual approaches can vary significantly depending on the order that clues are provided to the model. To alleviate this (and where feasible), we evaluate models across 16 random sortings of the clues. Due to cost, we did not evaluate GPT-3.5-turbo and GPT-4's sensitivity to this ordering; future work should report performance across multiple random sorts. Second, due to the nature of the quiz show *Only Connect*, the clues, groups, and connections in the dataset tend to be Western- (and specifically UK-) centric (e.g. "*Doctor Who companions*", "*English cricket captains*", "*Irish counties*"). Therefore, performance on the OCW dataset may not extrapolate to languages or cultures outside of Western English. In fact, the *US*-centric bias of LLMs like GPT-3.5/4 [84] might partially explain their poor performance on the *UK*-centric OCW dataset. We hope to add additional *Only Connect* inspired walls in multiple languages and with clues derived from various cultures & subcultures in future work. Finally, given that the walls are publicly available as text on fan sites like ocdb.cc, there is always the possibility that they are included in the training sets of LLMs like GPT. However, we think this is unlikely, given the low performance on the grouping and connection tasks. Preventing the test sets of publicly available datasets like our OCW from "leaking" into the training sets of LLMs remains an interesting and open problem. We have taken basic steps against this leakage by distributing our dataset in a compressed format [30].

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

# Appendices

## A  Additional Experiments

**Task 1 – Grouping**  In addition to grouping clue words using token embeddings (discussed in the main paper §4), we also ran grouping the words by clustering on 'contextual' embeddings. We experimentally induce 'context' by joining the sixteen (16) word tokens (in a random order) into a single pseudo-sentence. The embeddings for each token were different based on the ordering of the tokens. We repeat the random ordering sixteen times and report the mean and variance of the results obtained in Table 6.

| | WD ↓ | FMS ↑ | ARI ↑ | AMI ↑ | # Solved Walls | # Correct Groups |
|---|---|---|---|---|---|---|
| ELMo$_{LARGE}$ | $90.0 \pm .3$ | $23.6 \pm .4$ | $4.5 \pm .5$ | $5.6 \pm .7$ | $0 \pm 0$ | $19 \pm 3$ |
| DistilBERT$_{BASE}$ | $88.4 \pm .7$ | $26.7 \pm .3$ | $8.3 \pm .4$ | $10.4 \pm .5$ | $0 \pm 0$ | $30 \pm 4$ |
| BERT$_{LARGE}$ | $\mathbf{87.2 \pm .6}$ | $\mathbf{28.3 \pm .5}$ | $\mathbf{10.4 \pm .6}$ | $\mathbf{12.8 \pm .7}$ | $0 \pm 0$ | $\mathbf{46 \pm 5}$ |
| BERT$_{BASE}$ | $87.7 \pm .5$ | $28.0 \pm .2$ | $10.0 \pm .3$ | $12.4 \pm .4$ | $0 \pm 0$ | $39 \pm 2$ |
| RoBERTa$_{LARGE}$ | $88.4 \pm .5$ | $25.9 \pm .2$ | $7.4 \pm .3$ | $9.3 \pm .4$ | $0 \pm 0$ | $30 \pm 4$ |
| all-mpnet$_{BASE}$ | $87.6 \pm .5$ | $28.0 \pm .3$ | $10.0 \pm .4$ | $12.4 \pm .5$ | $0 \pm 0$ | $38 \pm 3$ |
| E5$_{LARGE}$ | $87.7 \pm .5$ | $28.1 \pm .3$ | $10.2 \pm .4$ | $12.7 \pm .5$ | $0 \pm 0$ | $37 \pm 4$ |
| E5$_{BASE}$ | $\mathbf{87.2 \pm .3}$ | $28.2 \pm .2$ | $10.2 \pm .3$ | $12.5 \pm .4$ | $0 \pm 0$ | $\mathbf{46 \pm 5}$ |
| Human Performace | – | – | – | – | 285 / 494 | 1405 / 1976 |

Table 6: Results of selected models on Task 1 (Grouping) using contextual embeddings. WD: Wasserstein Distance. FMS: Fowlkes Mallows Score. ARI: Adjusted Rand Index. NMI: Normalized Mutual Information. Mean $\pm$ standard deviation over 16 random seeds is shown. **Bold**: best scores.

**Task 2 – Connections**  In addition to prompting based results on GPT-4 (discussed in §4), we ran experiments on additional LLMs like LLaMa [67] (7B, 13B) using pre-trained configuration weights obtained by permission from Meta AI. However, without additional fine-tuning on the specific task, these LLMs were unable to solve the task in a meaningful manner. To elucidate, LLaMa generated a bunch of hallucinated words with unequal group sizes. We omit these unintelligible results for brevity.

# B    Additional Figures

In this section, we provide additional t-SNE projections of embeddings from various methods used.

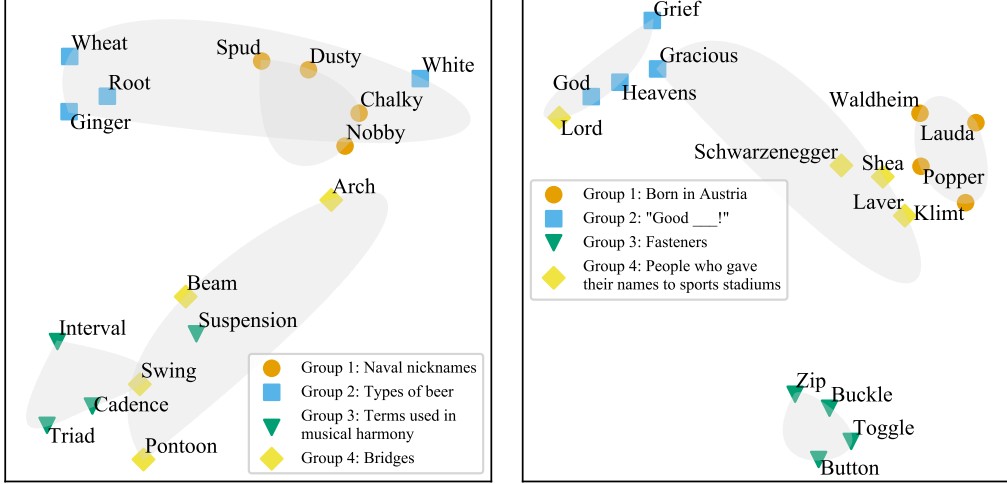

Figure 7: Solved wall for Task 1 (Grouping) using GloVe. **Left**: (`wall_id="7ed3"`), the embedding model erroneously associated the clue "*Suspension*" with the connection "*Bridges*"; however, this association is an example of a red herring. "*Suspension*" is "*a term used in musical harmony*" in this context. **Right**: (`wall_id="5e3c"`), shows that clue "*Lord*" is close to "*God, Heavens, and Grief*" in the embedding space, which matches the "*Good ___!*" connection. However, this is another example of a red herring as, in this context, "*Lord*" refers to "*Lord's cricket Ground*", a cricket stadium named after "*Thomas Lord*".

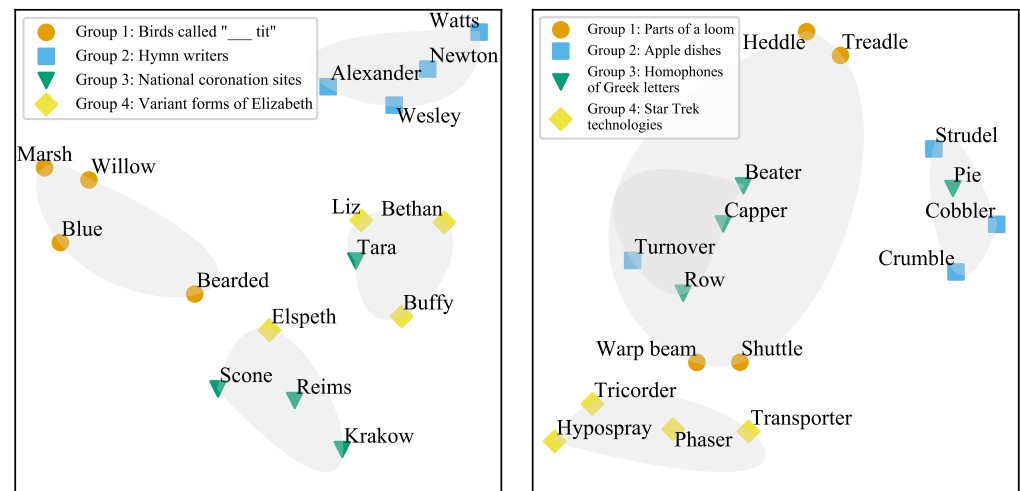

Figure 8: Solved wall for Task 1 (Grouping) using FastText (Crawl). **Left**: (`wall_id="d5e6"`), the embedding model erroneously associated the clue "*Tara*" other girls' names; but here, "*Tara*" is short for "*Hill of Tara*" and belongs to the "*national coronation sites*" group. **Right**: (`wall_id="4c22"`), shows that clue "*Pie*" associated with the connection "*Apple*". Even though it is acceptable in general context, here it represents a homophone for the Greek letter "π".

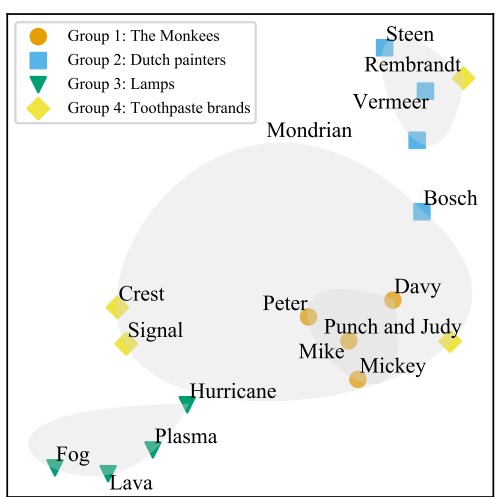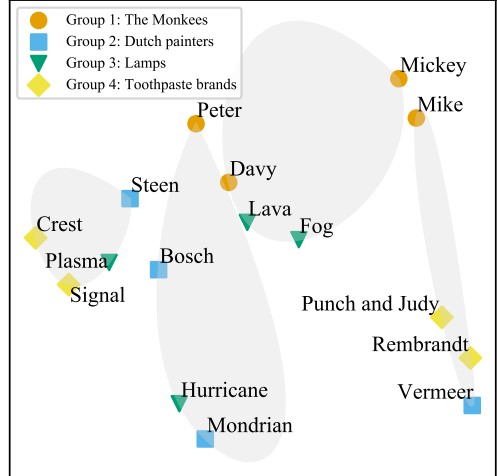

Figure 9: Solved wall (`wall_id="2d8f"`) for Task 1 (Grouping) using BERT$_{\text{LARGE}}$ with both static and contextual embeddings. **Left**: contextual embedding solved 3/4 groups. Here the clue "*Rambrandt*" is placed near other Dutch painters. The correct grouping for this clue in this wall is "*Toothpaste Brands*". **Right**: static embedding solved 0/4 groups.

# C Effects of Red-Herrings: Additional Experiments, Analysis and Results

## C.1 Additional Datasets

Both of the additional datasets described in this section for ablation experiments have been made available via our code repositories on Github and HuggingFace.

### C.1.1 OCW-Randomized Dataset

This test dataset generates a version of the test set where red herrings are removed or largely reduced in frequency. This is achieved by rebuilding every wall using a randomly selected group from different walls. We only applied the process to the (original OCW) test set, the train and validation sets are left untouched.

**Method**  For each wall in the existing test set, we leave the first group untouched, and sample three new groups, each from a different wall, such that none of the groups share a word in common. The connections for each group are unmodified. The result is a new version of the test set where every wall is composed of 4 random groups from 4 different walls.

### C.1.2 OCW-WordNet Dataset

WordNet [46, 20] is a large lexical database of English. Nouns, verbs, adjectives, and adverbs are grouped into sets of cognitive synonyms (synsets), each expressing a distinct concept. We use the hypernym/hyponym (or superlative/subordinative) hierarchical lexical structure aggregated in WordNet to generate an easy test set to further analyze the effects of red-herring in OCW.

**Method**  We use the existing words in a wall to select synonyms from the word's synsets. We only consider synsets that have at least five synonymous lexical names, then randomly sample four words. The original test set word and its definition (`ss.definition()`) subsequently becomes the connection phrase for the group. Four groups were generated for each wall, and the easy wall generation process was repeated for the total number of walls (494) in the original test data set.

For the group connections, we concatenate the superlative parent word with a synset definition giving a description of the word. This allows for an ideal semantic similarity score to be calculated using BERTScore. For a few cases (approx. 70/494 walls in the test set), the number of generated groups per wall is less than four, due to the unavailability of direct synonyms from word synsets. In those edge cases, we generate and append groups using common hypernym words like animal, mammal, furniture, etc. to ensure a wall is valid with four groups.

A sample generated easy group is shown below, where we prefix the group_id from the original OCW dataset with 'easy' to aid with mapping or identification.

```
{
    ...
    "group_3": {
    "group_id": "easy_691a_3",
    "gt_words": ["gibe","shaft","jibe","barb"],
    "gt_connection": "Shaft: an aggressive remark directed at a person
    like a missile and intended to have a telling effect"
    ...
}
```

Further, we generate easy to train and validation sets mimicking the original dataset, package and release these three additional easy sets, as **OCW-WordNet** as added contributions.

## C.2 Results of Ablation Experiments

### C.2.1 PLMs: Performance on Task 1 (Grouping)

We perform and present the results using 'static' embeddings due to the noted superior results and the word order related deficiency already shown by using contextual embeddings pertinent to our task setup.

| | WD $\downarrow$ | FMS $\uparrow$ | ARI $\uparrow$ | AMI $\uparrow$ | # Solved Walls | # Correct Groups |
|---|---|---|---|---|---|---|
| *Classic Word Embeddings* | | | | | | |
| GloVe | $76.8 \pm .7$ | $39.2 \pm .3$ | $24.0 \pm .4$ | $27.7 \pm .4$ | $7 \pm 1$ | $213 \pm 8$ |
| FastText (Crawl) | $76.1 \pm .5$ | $40.5 \pm .3$ | $25.0 \pm .6$ | $28.6 \pm .7$ | $\mathbf{13 \pm 1}$ | $236 \pm 7$ |
| FastText (News) | $79.3 \pm .5$ | $36.8 \pm .3$ | $21.0 \pm .3$ | $24.5 \pm .4$ | $5 \pm 1$ | $176 \pm 6$ |
| *Pre-trained Language Models (PLMs)* | | | | | | |
| ELMo$_{\text{LARGE}}$ | $80.9 \pm .4$ | $35.2 \pm .3$ | $18.9 \pm .3$ | $22.2 \pm .4$ | $3 \pm 1$ | $154 \pm 6$ |
| DistilBERT$_{\text{BASE}}$ | $82.3 \pm .6$ | $34.2 \pm .4$ | $17.7 \pm .5$ | $21.1 \pm .5$ | $1 \pm 1$ | $124 \pm 8$ |
| BERT$_{\text{LARGE}}$ | $86.2 \pm .4$ | $29.2 \pm .3$ | $11.5 \pm .3$ | $14.2 \pm .4$ | $0 \pm 0$ | $66 \pm 4$ |
| BERT$_{\text{BASE}}$ | $87.5 \pm .4$ | $27.7 \pm .3$ | $9.6 \pm .6$ | $11.8 \pm .5$ | $0 \pm 0$ | $48 \pm 4$ |
| RoBERTa$_{\text{LARGE}}$ | $86.7 \pm .5$ | $28.6 \pm .2$ | $10.8 \pm .3$ | $13.4 \pm .3$ | $1 \pm 0$ | $56 \pm 4$ |
| *Sentence Transformers* | | | | | | |
| all-mpnet$_{\text{BASE}}$ | $81.4 \pm .4$ | $35.1 \pm .4$ | $18.9 \pm .5$ | $22.0 \pm .6$ | $8 \pm 1$ | $154 \pm 7$ |
| E5$_{\text{LARGE}}$ | $76.0 \pm .5$ | $40.7 \pm .3$ | $25.9 \pm .4$ | $29.7 \pm .4$ | $8 \pm 1$ | $230 \pm 5$ |
| E5$_{\text{BASE}}$ | $\mathbf{75.1 \pm .8}$ | $\mathbf{41.8 \pm .3}$ | $\mathbf{27.2 \pm .3}$ | $\mathbf{31.1 \pm .3}$ | $8 \pm 1$ | $\mathbf{249 \pm 8}$ |
| Human Performance | – | – | – | – | – | – |

Table 7: Results of **OCW-Randomized** using static embeddings. WD: Wasserstein Distance. FMS: Fowlkes Mallows Score. ARI: Adjusted Rand Index. NMI: Normalized Mutual Information. Mean $\pm$ standard deviation over 16 random seeds is shown. **Bold**: best scores.

| | WD $\downarrow$ | FMS $\uparrow$ | ARI $\uparrow$ | AMI $\uparrow$ | # Solved Walls | # Correct Groups |
|---|---|---|---|---|---|---|
| *Classic Word Embeddings* | | | | | | |
| GloVe | $43.0 \pm 1.0$ | $66.1 \pm .4$ | $57.4 \pm .5$ | $60.9 \pm .5$ | $118 \pm 3$ | $886 \pm 1$ |
| FastText (Crawl) | $30.6 \pm 1.0$ | $75.8 \pm .6$ | $69.6 \pm .7$ | $72.4 \pm .7$ | $195 \pm 6$ | $1173 \pm 18$ |
| FastText (News) | $44.9 \pm 1.2$ | $64.9 \pm .5$ | $55.9 \pm .6$ | $59.5 \pm .6$ | $105 \pm 3$ | $844 \pm 12$ |
| *Pre-trained Language Models (PLMs)* | | | | | | |
| ELMo$_{\text{LARGE}}$ | $52.5 \pm 1.1$ | $58.9 \pm .3$ | $48.2 \pm .4$ | $52.5 \pm .4$ | $67 \pm 3$ | $682 \pm 9$ |
| DistilBERT$_{\text{BASE}}$ | $45.5 \pm 1.0$ | $64.1 \pm .4$ | $55.0 \pm .5$ | $58.7 \pm .5$ | $105 \pm 3$ | $835 \pm 13$ |
| BERT$_{\text{LARGE}}$ | $76.9 \pm 1.0$ | $38.9 \pm .2$ | $23.4 \pm .3$ | $27.5 \pm .3$ | $7 \pm 0$ | $197 \pm 6$ |
| BERT$_{\text{BASE}}$ | $73.0 \pm 1.3$ | $42.5 \pm .5$ | $27.9 \pm .6$ | $32.5 \pm .6$ | $8 \pm 2$ | $268 \pm 12$ |
| RoBERTa$_{\text{LARGE}}$ | $57.4 \pm 1.3$ | $54.8 \pm .3$ | $43.3 \pm .3$ | $47.5 \pm .3$ | $48 \pm 2$ | $573 \pm 8$ |
| *Sentence Transformers* | | | | | | |
| all-mpnet$_{\text{BASE}}$ | $\mathbf{22.6 \pm .7}$ | $\mathbf{81.9 \pm .4}$ | $\mathbf{77.1 \pm .5}$ | $\mathbf{79.4 \pm .4}$ | $\mathbf{256 \pm 4}$ | $\mathbf{1365 \pm 12}$ |
| E5$_{\text{LARGE}}$ | $23.6 \pm .8$ | $80.9 \pm .4$ | $75.9 \pm .5$ | $78.3 \pm .4$ | $250 \pm 4$ | $1347 \pm 12$ |
| E5$_{\text{BASE}}$ | $26.9 \pm .9$ | $78.0 \pm .4$ | $72.3 \pm .5$ | $75.0 \pm .5$ | $224 \pm 4$ | $1259 \pm 10$ |
| Human Performance | – | – | – | – | – | – |

Table 8: Results of **OCW-WordNet** using static embeddings. WD: Wasserstein Distance. FMS: Fowlkes Mallows Score. ARI: Adjusted Rand Index. NMI: Normalized Mutual Information. Mean $\pm$ standard deviation over 16 random seeds is shown. **Bold**: best scores.

### C.2.2 LLMs: Performance on Task 1 (Grouping) using GPT3.5/4

Here we present the results of repeating Task 1 (grouping) on the ablation datasets OCW-Randomized (C.1.1) and OCW-Wordnet (C.1.2) to analyze the effects of red-herrings in walls on LLM performance.

The results adhere to the expected results of superior performance with the dilution/removal of red-herrings from the walls.

| | # In-context Examples | WD ↓ | FMS ↑ | ARI ↑ | AMI ↑ | # Solved Walls | # Correct Groups |
|---|---|---|---|---|---|---|---|
| GPT-3.5-turbo | 0-shot | 74.3 | 40.4 | 26.4 | 29.8 | 5 | 274 |
| | 1-shot | 72.0 | 43.1 | 29.0 | 32.3 | 12 | 315 |
| | 3-shot | 72.7 | 43.4 | 29.4 | 32.9 | 10 | 306 |
| | 5-shot | 70.7 | 44.6 | 30.9 | 34.4 | 16 | 337 |
| | 10-shot | 70.5 | 43.8 | 30.0 | 33.5 | 17 | 333 |
| GPT-4 | 0-shot | 58.2 | 56.2 | 45.4 | 48.8 | 59 | 595 |
| | 1-shot | 55.1 | **58.0** | **47.5** | **51.0** | 57 | 644 |
| | 3-shot | 55.0 | 57.5 | 46.9 | 50.3 | 62 | 649 |
| | 5-shot | **54.1** | **58.0** | **47.5** | 50.9 | **68** | **655** |
| | 10-shot | 56.6 | 56.1 | 45.1 | 48.5 | 55 | 614 |
| Human Performance | | – | – | – | – | – | – |

Table 9: Results of **OCW-Randomized** using Large Language Models. WD: Wasserstein Distance. FMS: Fowlkes Mallows Score. ARI: Adjusted Rand Index. NMI: Normalized Mutual Information. **Bold**: best scores.

| | # In-context Examples | WD ↓ | FMS ↑ | ARI ↑ | AMI ↑ | # Solved Walls | # Correct Groups |
|---|---|---|---|---|---|---|---|
| GPT-3.5-turbo | 0-shot | 15.9 | 86.3 | 83.4 | 84.9 | 337 | 1522 |
| | 1-shot | 24.8 | 76.4 | 74.4 | 75.4 | 320 | 1400 |
| | 3-shot | 8.65 | 92.7 | 91.2 | 91.8 | 415 | 1748 |
| | 5-shot | 8.09 | 94.0 | 92.4 | 93.1 | 415 | 1759 |
| | 10-shot | 6.55 | 95.3 | 94.0 | 94.7 | 428 | 1800 |
| GPT-4 | 0-shot | **1.51** | **98.5** | **98.0** | **98.2** | **471** | **1926** |
| | 1-shot | 19.2 | 87.9 | 84.3 | 83.7 | 304 | 1581 |
| | 3-shot | 21.5 | 86.6 | 82.5 | 81.8 | 279 | 1537 |
| | 5-shot | 19.1 | 88.1 | 84.5 | 83.8 | 298 | 1584 |
| | 10-shot | 11.2 | 92.9 | 90.7 | 90.4 | 378 | 1742 |
| Human Performance | | – | – | – | – | – | – |

Table 10: Results of **OCW-WordNet** using Large Language Models. WD: Wasserstein Distance. FMS: Fowlkes Mallows Score. ARI: Adjusted Rand Index. NMI: Normalized Mutual Information. **Bold**: best scores.

