# Appendices

## A   Additional Experiments

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

## C  Datasheet

The following section provides answers to questions listed in datasheets for datasets.

---

### MOTIVATION

---

**For what purpose was the dataset created?**  Was there a specific task in mind? Was there a specific gap that needed to be filled? Please provide a description.

The OCW dataset is created to be an analogical proxy for the Remote Associates Test (RAT) [45] from cognitive neuroscience in evaluating LLMs for human-imitative *creative problem-solving*. The presented clues have heterogeneous connections with open-domain knowledge retrieval and contain red herrings or misleading stimuli by design. The two tasks entails *grouping* sixteen (16) jumbled up clue words into associated groups, and naming the right *connection* for each group. To the best of our knowledge, there are no existing tasks for evaluating LLMs for human-like creative problem solving in existing, and concurrent benchmarks including the BIG-Bench, HELM, Global-Bench. Thus, this dataset and tasks are valuable additions for overall LLM evaluation and measuring progress towards human-imitative AI.

**Who created this dataset (e.g., which team, research group) and on behalf of which entity (e.g., company, institution, organization)?**
The dataset has been collectively curated by the authors of this paper.

**What support was needed to make this dataset?**  (e.g.who funded the creation of the dataset? If there is an associated grant, provide the name of the grantor and the grant name and number, or if it was supported by a company or government agency, give those details.)
This work was supported by Natural Sciences and Engineering Research Council of Canada (NSERC).

---

### COMPOSITION

---

**What do the instances that comprise the dataset represent (e.g., documents, photos, people, countries)?**  Are there multiple types of instances (e.g., movies, users, and ratings; people and interactions between them; nodes and edges)? Please provide a description.
Each instance contains a connecting wall puzzle and its solution from the popular quiz show Only Connect.

**How many instances are there in total (of each type, if appropriate)?**
618 wall puzzles (instances of the dataset), for a total of 2,472 groups, and 9,888 clues.

**Does the dataset contain all possible instances or is it a sample (not necessarily random) of instances from a larger set?**
The dataset has been curated from the first fifteen seasons of the "Only Connect" show, which accounts for approximately 81% of the total seasons. The latest season, Season 18, was concluded in March 2023.

**What data does each instance consist of?**  "Raw" data (e.g., unprocessed text or images) or features? In either case, please provide a description.
Each instance contains a connecting wall puzzle and its clues and solution. All instances are in English and provided as text strings in JSON format.

---

**Is there a label or target associated with each instance?** If so, please provide a description.
Yes. The labels for Task 1 are the solved walls, and for Task 2 the ground-truth connections.

**Is any information missing from individual instances?** If so, please provide a description, explaining why this information is missing (e.g., because it was unavailable). This does not include intentionally removed information, but might include, e.g., redacted text.
N/A

**Are relationships between individual instances made explicit (e.g., users' movie ratings, social network links)?** If so, please describe how these relationships are made explicit.
Each wall is given a unique ID. Clues and solutions associated with each wall belong to the same JSON object as that wall.

**Are there recommended data splits (e.g., training, development/validation, testing)?** If so, please provide a description of these splits, explaining the rationale behind them.
We randomly split the dataset into the training/dev/test set according to a proportion of 1:1:8. The primary goal of our dataset is to evaluate the zero- and few-shot creative problem-solving abilities of Large Language Models; as such, we elect to set the size of the test set to be much greater than train or validation sets.

**Are there any errors, sources of noise, or redundancies in the dataset?** If so, please provide a description.
The dataset has undergone a thorough review and is subjected to both automated and manual checks as part of a strict quality control protocol.

**Is the dataset self-contained, or does it link to or otherwise rely on external resources (e.g., websites, tweets, other datasets)?**
The dataset is self-contained.

**Does the dataset contain data that might be considered confidential (e.g., data that is protected by legal privilege or by doctor-patient confidentiality, data that includes the content of individuals' non-public communications)?** If so, please provide a description.
N/A.

**Does the dataset contain data that, if viewed directly, might be offensive, insulting, threatening, or might otherwise cause anxiety?** If so, please describe why.
N/A.

**Does the dataset relate to people?** If not, you may skip the remaining questions in this section.
The dataset does not have individual-specific information.

## COLLECTION

**How was the data associated with each instance acquired?** Was the data directly observable (e.g., raw text, movie ratings), reported by subjects (e.g., survey responses), or indirectly inferred/derived from other data (e.g., part-of-speech tags, model-based guesses for age or language)? If data was reported by subjects or indirectly inferred/derived from other data, was the data validated/verified? If so, please describe how.
The wall puzzles were scraped from the fan website ocdb.cc as well as manually watching the episodes. Human performance results were manually curated from the episodes. all data verified

through manual watching of episodes.

**What mechanisms or procedures were used to collect the data (e.g., hardware apparatus or sensor, manual human curation, software program, software API)?** How were these mechanisms or procedures validated?
We utilized python's BeautifulSoup library to scrape only connect fan websites. all episodes were watched manually for human performance collection, and the same procedure validated the data collection.

**What was the resource cost of collecting the data?** (e.g. what were the required computational resources, and the associated financial costs
Experiments were run using NVIDIA GeForce RTX 2080 Ti GPU system.

**Who was involved in the data collection process (e.g., students, crowdworkers, contractors) and how were they compensated (e.g., how much were crowdworkers paid)?**
The authors of this paper.

**Were any ethical review processes conducted (e.g., by an institutional review board)?** If so, please provide a description of these review processes, including the outcomes, as well as a link or other access point to any supporting documentation.
N/A.

**Does the dataset relate to people?** If not, you may skip the remainder of the questions in this section.
The dataset does not have individual-specific information.

## PREPROCESSING / CLEANING / LABELING

**Was any preprocessing/cleaning/labeling of the data done(e.g.,discretization or bucketing, tokenization, part-of-speech tagging, SIFT feature extraction, removal of instances, processing of missing values)?** If so, please provide a description. If not, you may skip the remainder of the questions in this section.
N/A.

**Was the "raw" data saved in addition to the preprocessed/cleaned/labeled data (e.g., to support unanticipated future uses)?** If so, please provide a link or other access point to the "raw" data.
N/A.

**Is the software used to preprocess/clean/label the instances available?** If so, please provide a link or other access point.
N/A.

## USES

**Has the dataset been used for any tasks already?** If so, please provide a description.
No.

**Is there a repository that links to any or all papers or systems that use the dataset?** If so, please provide a link or other access point.

No.

**What (other) tasks could the dataset be used for?**
Evaluation of Large Language Models for creative problem-solving as well as Artificial General Intelligence tasks.

**Is there anything about the composition of the dataset or the way it was collected and preprocessed/cleaned/labeled that might impact future uses?** For example, is there anything that a future user might need to know to avoid uses that could result in unfair treatment of individuals or groups (e.g., stereotyping, quality of service issues) or other undesirable harms (e.g., financial harms, legal risks) If so, please provide a description. Is there anything a future user could do to mitigate these undesirable harms?
N/A.

**Are there tasks for which the dataset should not be used?** If so, please provide a description.
We caution regarding unethical reuse of the dataset, specifically for the purpose of training future reasoning engines for unethical use cases.

---

## DISTRIBUTION

**Will the dataset be distributed to third parties outside of the entity (e.g., company, institution, organization) on behalf of which the dataset was created?** If so, please provide a description.
No.

**How will the dataset will be distributed (e.g., tarball on website, API, GitHub)?** Does the dataset have a digital object identifier (DOI)?
The code and link to the dataset is available at `https://github.com/TaatiTeam/OCW`

**When will the dataset be distributed?**
Now.

**Will the dataset be distributed under a copyright or other intellectual property (IP) license, and/or under applicable terms of use (ToU)?** If so, please describe this license and/or ToU, and provide a link or other access point to, or otherwise reproduce, any relevant licensing terms or ToU, as well as any fees associated with these restrictions.
The dataset is released under MIT License.

**Have any third parties imposed IP-based or other restrictions on the data associated with the instances?** If so, please describe these restrictions, and provide a link or other access point to, or otherwise reproduce, any relevant licensing terms, as well as any fees associated with these restrictions.
No.

**Do any export controls or other regulatory restrictions apply to the dataset or to individual instances?** If so, please describe these restrictions, and provide a link or other access point to, or otherwise reproduce, any supporting documentation.
No.

**Who is supporting/hosting/maintaining the dataset?**

The dataset is hosted on University of Toronto Computer Science Department servers and will be maintained by the authors of this paper.

**How can the owner/curator/manager of the dataset be contacted (e.g., email address)?**

The maintainers can be contacted via email: saeid.alavi@mail.utoronto.ca, raeidsaqur@cs.toronto.edu, john.giorgi@mail.utoronto.ca, mozhgans@stanford.edu, babak.taati@uhn.ca.

**Is there an erratum?** If so, please provide a link or other access point.

No.

**Will the dataset be updated (e.g., to correct labeling errors, add new instances, delete instances)?** If so, please describe how often, by whom, and how updates will be communicated to users (e.g., mailing list, GitHub)?

The authors plan to continue updating the dataset, including but not limited to scaling the dataset to include more seasons, providing new test/dev sets, and organizing shared tasks with the dataset. The updates will be yearly and communicated to users through public shared tasks.

**If the dataset relates to people, are there applicable limits on the retention of the data associated with the instances (e.g., were individuals in question told that their data would be retained for a fixed period of time and then deleted)?** If so, please describe these limits and explain how they will be enforced.

N/A.

**Will older versions of the dataset continue to be supported/hosted/maintained?** If so, please describe how. If not, please describe how its obsolescence will be communicated to users.

Yes, the authors are committed to maintaining and updating the older versions of the dataset.

**If others want to extend/augment/build on/contribute to the dataset, is there a mechanism for them to do so?** If so, please provide a description. Will these contributions be validated/verified? If so, please describe how. If not, why not? Is there a process for communicating/distributing these contributions to other users? If so, please provide a description.

Any potential contributors are welcome to expand the dataset to larger size through contacting the authors of the paper.