# OpenReview forum: "Large Language Models are Fixated by Red Herrings: Exploring Creative Problem Solving and Einstellung Effect using the Only Connect Wall Dataset"
_NeurIPS.cc/2023/Track/Datasets_and_Benchmarks — NeurIPS 2023 Datasets and Benchmarks Poster_

### Official Review · Reviewer_9y7c · 2023-07-19
**Review of Submission #231: Large Language Models are Fixated by Red Herrings: Exploring Creative Problem Solving and Einstellung Effect using the Only Connect Wall Dataset**

**Rating:** 6
**Confidence:** 3

**Strengths:**

The authors have done an impressive amount of work augmenting an existing dataset (ocdb.cc - i.e. Anotado, Ruddle & Halbur (2023)) with other data from watching the actual show itself to fill in incomplete gaps. The authors have also managed to operationalise an LLM-based and an embedding-based clustering benchmark on a structured RAT problem.

**Additional Feedback:**

A very well-written piece with a few improvements that the reviewer hopes the authors can incorporate when preparing their piece for further consideration. (as a side note: The dataset also has the potential to benefit the general public who want to learn something new or to practice problem-solving and creative thinking skills, the same way the NYT Connections game has brought OC to its general readership). The reviewer wishes the authors every success.

**Clarity:**

Yes, the paper is well written. (Small hints for improvement were covered in *Limitations* above).
Please note that ocdb.cc can be cited as Anotado, Ruddle & Halbur (2023) - see https://ocdb.cc/staff/
Also, the production companies behind Only Connect need to be cited. See *Documentation*.

**Correctness:**

The reviewer commends the authors for putting lots of effort in to verify the correctness of the wall data from Anotado, Ruddle & Halbur (2023). There are some gaps (not an issue of Correctness per se) identified in *Limitations* that hopefully the authors can bear in mind.

**Documentation:**

The reviewer commends the authors for producing a Datasheet for the dataset inspired by Gebru et al, (2018, 2021).

Some issues to address (p.22 of supplement): "The wall puzzles were scraped from fan websites, and human performance results were manually... curated from the episodes. all data verified through manual watching of episodes."
- please clarify if fan websites (multiple i.e. plural) were used - in which case, please cite **all** sources.

Also in the Checklist, #4(d) "no information from participants on the show (besides anonymized and aggregated performance statistics) was collected" -- a recommendation to the authors would be to disambiguate "anonymized and aggregated performance statistics" and clearly state that the statistics pertain to the statuses of the CWs (solved/unsolved) rather than the players.

Finally, per Checklist #4(a) - "(a) If your work uses existing assets, did you cite the creators? [Yes]" please attribute the actual curators of ocdb.cc -- a point repeated in prior discussion above. Also take note to cite the work of the Only Connect producers and production teams -- https://guides.library.uq.edu.au/referencing/apa7/television -- as they are ultimately the point of credit for the original Walls.



**Ethics:**

One small comment is maybe for the authors to consider, and future proof against: any out-of-scope use, unsupported use cases, and unethical reuse of data in the datasheet (e.g., training future reasoning engines for unethical use cases) -- see Checklist #1(c) "Did you discuss any potential negative social impacts of your work? [No]"

**Limitations:**

There are two limitations as previously alluded to:

1. LLMs might exhibit USA-centric bias, and Western-bias in general. To the authors' credit, they have acknowledged this, as:
> Therefore, performance on the OCW dataset may not extrapolate to language or subcultures outside of Western English. We hope to add additional Only Connect inspired walls in multiple languages and with clues derived from various subcultures in future work.

Also, this may pose an additional [technical] limitation: per Wolfe & Caliskan (2022) (https://arxiv.org/pdf/2207.00691.pdf) an LLM which could hypothetically contain implicit bias towards, say, US-based content, might not translate well to performance on a UK-based quiz show. The authors might need to expand on this a bit more.

* also a side note: the authors might want to add "**cultures**" to "...extrapolate to language or subcultures" -- it is not only subcultures (i.e. as defined by the Cambridge Dictionary as "...group of people within a society that are different from the rest of that society"), but also different cultures more broadly.

2. The JSON file is, in the main, self-documenting. However, there might be the need for additional information, as not all are readily clear especially to a non-OC viewer (disclaimer: this reviewer watches OC quite regularly but had to do a double-take when studying the JSON).
* For example: describing the flow of OC's CW round -> grouping attempts -> completion OR timeout OR 3-attempts-lost -> connection determination.
* In addition, the case of: {timeout OR 3-attempts-lost scenario AND successful connections post-grouping by the OC's presenter} might need to be treated differently to a case of {a team not knowing the grouping at all AND failed connections even after reveal due to a lack of, say, domain expertise on an esoteric topic}
* It is beneficial to add the year (month?) to the season data, as it allows one to gauge the performance of, say, different releases of LLMs with respect to the chronological context of a set of clues (in terms of season).

**Opportunities For Improvement:**

It is pertinent to note that there are *Limitations* to the evaluation process (which to the authors' credit, have been acknowledged). They will be detailed in *Limitations* below, but here is a summary at a glance: (1) LLMs might exhibit USA-centric bias (see Wolfe & Caliskan (2022) https://arxiv.org/pdf/2207.00691.pdf) which might not translate to performance on a UK dataset. (2) The parameters in the JSON - including most importantly date and performance - need clarification/qualification.

Note other suggestions in *Clarity* below.

**Relation To Prior Work:**

The literature review was quite comprehensive in this reviewer's opinion.

**Summary And Contributions:**

The contribution is summarised as follows: the authors have presented a curated, structured collection of Connecting Walls (CWs) from the Only Connect (OC) quiz show with N=618 (exceeding the extant largest of N=369 by Anotado, Ruddle & Halbur (2023), i.e. the "OCDB"). These CWs also come with ground truth answers and ground truth performances (i.e., if the human expert contestants on OC have solved them or otherwise).

The dataset is complemented by "evaluation of selected pre-trained language models and LLMs ... on creative problem solving tasks... and [more importantly, in the reviewer's perspective] identifying correct open knowledge domain connections in respective groups".


* EDIT for transparency: revised Confidence Score, misclicked on the wrong Confidence Rating initially, with apologies from this reviewer.

---

> ### Author Response · Authors · 2023-08-18
> **Reviewer 9y7c response: Revised dataset and manuscript with suggested improvements**
>
> We thank the reviewer for their insightful comments and suggestions. Our answers to your comments and questions are as follows.
>
> >  this may pose an additional [technical] limitation: per Wolfe & Caliskan (2022) (https://arxiv.org/pdf/2207.00691.pdf) an LLM which could > hypothetically contain implicit bias towards, say, US-based content, might not translate well to performance on a UK-based quiz show.
> > The authors might need to expand on this a bit more.
> >* also a side note: the authors might want to add "cultures" to "...extrapolate to language or subcultures" -- it is not only subcultures (i.e.
> > as defined by the Cambridge Dictionary as "...group of people within a society that are different from the rest of that society"), but also
> > different cultures more broadly.
>
> We appreciate the great suggestion. In the revised **Limitations** section, we added 'cultures' and expanded on the limitations caused by potential US-biased models (and added the suggested reference).
>
> > please clarify if fan websites (multiple i.e. plural) were used - in which case, please cite all sources.
>
> > per Checklist #4(a) - "(a) If your work uses existing assets, did you cite the creators? [Yes]" please attribute the actual curators of
> > ocdb.cc -- a point repeated in prior discussion above. Also take note to cite the work of the Only Connect producers and production teams -> - https://guides.library.uq.edu.au/referencing/apa7/television -- as they are ultimately the point of credit for the original Walls.
>
> We thank the reviewer for the good suggestion. The wall puzzles were scraped from the fan website ocdb.cc as well as manually watching the episodes. We have clarified the curation resources in the appendix (p.23) and cited the ocdb fan website (p.3 footnote). We have also cited the Only Connect producers and production teams (p. 3). Additionally, we have added an acknowledgement section on both the GitHub page and dataset README and thanked both the curators of the fan website and producers/host of the Only Connect quiz show.
>
> > in the Checklist, #4(d) "no information from participants on the show (besides anonymized and aggregated performance statistics) was > collected" -- a recommendation to the authors would be to disambiguate "anonymized and aggregated performance statistics" and
> > clearly state that the statistics pertain to the statuses of the CWs (solved/unsolved) rather than the players.
>
> Thank you for the suggestion. We have provided further clarification in checklist #4(d) by specifying that the statistics refer to the scores of the CWs (solved/partially solved/unsolved) rather than the players.
>
> > The JSON file is, in the main, self-documenting. However, there might be the need for additional information, as not all are readily clear > especially to a non-OC viewer (disclaimer: this reviewer watches OC quite regularly but had to do a double-take when studying the
> > JSON).
>
> Great suggestion. We have completely revamped our GitHub README to clearly explain each element of the datasets structure, and we now distribute this README with copies of the dataset.
>
> > It is beneficial to add the year (month?) to the season data, as it allows one to gauge the performance of, say, different releases of LLMs > with respect to the chronological context of a set of clues (in terms of season).
>
> Yes, we believe it is and thanks for the suggestion. We have added the start and end date of each season to the dataset files as a value for **season_to_walls_map** key and updated the README accordingly.
>
> > One small comment is maybe for the authors to consider, and future proof against: any out-of-scope use, unsupported use cases, and
> > unethical reuse of data in the datasheet (e.g., training future reasoning engines for unethical use cases) -- see Checklist #1(c) "Did you > discuss any potential negative social impacts of your work? [No]"
>
> Many thanks for this ethics comment. We have set the Checklist #1(c) flag as [Yes] and provided the caution regarding unethical reuse of the dataset in section 5.5 of the datasheet.

---

### Official Review · Reviewer_H46b · 2023-07-21
**A challenging task for LLMs; Missing analysis of red herrings**

**Rating:** 6
**Confidence:** 4

**Strengths:**

1. Propose a challenging task, word grouping with heterogeneous connections, that requires creative problem solving and association and build the evaluation dataset.
2. Provide various baseline results including clustering algorithms and in-context learning of LLMs, along with the human performance baseline.

**Additional Feedback:**

None

**Clarity:**

LLMs are used for both task 1 (grouping) and task 2 (connection), but it is not listed as the method in section 3.1.

**Correctness:**

The paper claims that the puzzles in the quiz show contain built-in, deliberate red herrings. More statistics, such as ratio of problems containing red herrings and types of red herrings are required to support the claim.

**Documentation:**

Yes

**Limitations:**

See opportunities for improvement.

**Opportunities For Improvement:**

The title of the paper emphasizes "red herrings" and "Einstellung Effect". However, analysis of the distractors and their influence on the model performance is missing. The paper assumes that the puzzles in the quiz show contain built-in, deliberate red herrings. More statistics, such as ratio of problems containing red herrings and types of red herrings are required to support the claim. Also, the paper should provide analysis of influence of red herrings on the model performance. For example, how many incorrect examples can be attributed to misclassification of red herrings? Will the model performance improve if replacing red herrings with normal words?

**Relation To Prior Work:**

Yes

**Summary And Contributions:**

This paper proposes a challenging task for LLMs: word grouping with heterogeneous connections. The paper builds an evaluation dataset for the task from the quiz show Only Connect. Various baselines, including clustering algorithms based on static and contextualized word embeddings and in-context learning based on LLMs, along with human performance results from the quiz show. The results show that SOTA LLMs perform significantly worse than human performance.

---

> ### Author Response · Authors · 2023-08-18
> **Reviewer H46b response: Revised manuscript specifically addresses analysis of red herrings on model performance**
>
> We thank you for the insightful comments and valuable suggestions to improve our contributions.
>
> > However, analysis of the distractors and their influence on the model performance is missing.
>
> Adding analysis of red-herrings (distractors) and their influence on our results is a great suggestion that was echoed by other reviewers as well. As outlined in our *global response*, we added **section 4.3** to answer these questions with additional empirical results. Specifically, our second test dataset, **OCW-WordNet**, was inspired by this reviewer's question: *Will the model performance improve if replacing red herrings with normal words?* (Yes, it does). We are hopeful that the added empirical results using two additional datasets (with varying, controlled amount of red herrings) will satisfy the reviewer's well-received suggestions of improvement.
>
> > More statistics, such as ratio of problems containing red herrings and types of red herrings are required to support the claim.
>
> *Correctness*: We further clarify that each wall of the OCW dataset (in all sets) contain deliberately designed red herrings by construction. So, the ratio of problems containing red herrings is moot (100%). As for the **type** of distractors, there is no categorization/delineation of distractors in the designed walls. Please see the 2nd paragraph of the Introduction section (ln 39-51) where we explain the many possibilities of having heterogenous (e.g., synonymy, semantic, compounding) associations, and degrees of figurativeness or abstractness among words. Compiling, then categorizing red herrings present in every wall into these connections buckets is beyond the scope of our work.
>
> > LLMs are used for both task 1 (grouping) and task 2 (connection), but it is not listed as the method in section 3.1.
>
> *Clarity* The absence of Task 1 (grouping) in section 3.1 is an astute observation. This is indeed the case, because we only show the efficacy of pre-trained embeddings in section 3.1 using off-the-shelf embeddings (GLoVE) and PLMs (e.g. BERT), that does not allow solving for Task 2 (connections). We show Task 2 (connections) via prompting using LLMs only.

---

> > ### Comment · Reviewer_H46b · 2023-08-26
> >
> > Changed my rating to 6

---

### Official Review · Reviewer_zJGy · 2023-07-21
**Interesting new task**

**Rating:** 7
**Confidence:** 4
**Correctness:** Yes
**Clarity:** I find the paper overall well written…

**Strengths:**

The task is interesting, and evaluations on the state of models show that they are challenging. Therefore it should give headroom to improve the capabilities needed for these sorts of tasks.

The evaluations are comprehensive and easy to follow. I especially appreciate the inclusion of more classic methods using clustering and embeddings, instead of pure prompting.

The framing of the red herring is very interesting and thought provoking.


**Additional Feedback:**

N/A

**Documentation:**

Yes

**Ethics:**

Curious if the Only Connect show is copyrighted and if that creates a legal complication.

**Limitations:**

Yes

**Opportunities For Improvement:**

I find the framing of ‘human-imitative behavior’ or ‘human-like behavior’ in the abstract a bit disturbing. But maybe it’s just a matter of taste. The task can be introduced without focusing too much on the human-like behavior and can still be valuable as a complex and challenging task.

The framing of the fixation effect, while interesting, is not sufficiently supported by empirical evidence. A proper way to evaluate whether language models are fixed by red herrings would be to have two splits of the same task, one with red herring, one without, and compare their performance. Some additional analysis of the language model outputs is also necessary to validate that the “red herrings” are indeed the reason for failure. I didn’t find these analyses in the paper, so I would recommend toning down the claims about how language models are fixed by red herrings, moving to discussions as a framing device rather than a main result.


**Relation To Prior Work:**

Yes

**Summary And Contributions:**

The paper creates a new task / dataset for creative problem solving – Only Connect Wall. Evaluation results with pre-trained language models are also reported, as well as more classic methods such as clustering. The new task is framed in the context of red herrings and fixation effect and Einstellung paradigm.

Overall I find the paper novel, well written, and could be a valuable contribution to the community focused on creative problem solving abilities.

---

> ### Author Response · Authors · 2023-08-18
> **Reviewer zJGy response: Revised manuscript with suggested improvements, and addressed ethics comment**
>
> We thank the reviewer for the positive feedback and valuable suggestions. We respond to the two main points raised by the reviewer below:
>
> > The framing of the fixation effect, while interesting, is not sufficiently supported by empirical evidence. A proper way to evaluate whether
> > language models are fixed by red herrings would be to have two splits of the same task, one with red herring, one without, and compare
> > their performance.
>
> We thank the reviewer for the suggestion; other reviewers echoed similar suggestions. To directly address this concern, we constructed two new versions of the dataset: __OCW-Randomized__ (with reduced red herrings) and __OCW-WordNet__ (with zero red herrings) and presented the results in the new Table 5. Please see the __global response__ for more details. The results provide further evidence for our hypothesis that LLMs suffer from the fixation effect.
>
> > Curious if the Only Connect show is copyrighted and if that creates a legal complication.
>
> To the best of our knowledge, this dataset has no legal restrictions, as all information compiled is publicly and freely available. Therefore, the basic template MIT license (in the dataset's repository) is appropriate. We have added citations for both the Only Connect quiz show host and production team as well as the fan website (ocdb.cc) in the revised manuscript (page 3) and dataset README (acknowledgement section). Lastly, we note that we made sure scraping of ocdb.cc was allowed by referring to the robots.txt file.

---

### Official Review · Reviewer_fDpy · 2023-07-25
**interesting new direction**

**Rating:** 6
**Confidence:** 3
**Correctness:** yes
**Clarity:** yes

**Strengths:**

- the cognitive motivation to study creative problem solving (, fixation effect, Einstellung paradigm, etc.) is very interesting and reasonable, and novel in the NLP community.

- experiments (baseline design, task setup, metrics, results) are generally reasonable, clear,  comprehensive, and helpful for future research in this direction. some concrete findings/insights:
1. contextual embeddings do not perform well for RAT-style tasks, due to e.g. format mismatch
2. more in-context examples do not improve performance for such "creative" tasks
3. still space for improvement to humans

**Additional Feedback:**

I was wondering if LLM might have memorized some of the examples (given there's only <700 puzzles), and some validation can be done. but seems less concerning given the performance is not great.

**Documentation:**

yes

**Limitations:**

authors note limitations of variance (for contextual embeddings) and language (current data based on UK show), see more above.

**Opportunities For Improvement:**

would be happy to raise score if some can be addressed.

- better evaluation around fixation effect & Einstellung paradigm: the experiments and findings feel a bit short given all the cognitive motivations stacked in abstract/intro. given data collection is relatively easy in this work, some more insightful evaluation and analysis could've been conducted to strengthen the work. for example, the finding that "more few-shot examples for LLM does not improve or even slightly hurt grouping" may be strengthened by some error bar and variance (Table 4). it'd also be nice and interesting to sort puzzles by human difficulty/distraction type and compare to some model performance. in sum, only very basic and aggregated results are shown with simple hyper parameters (#shots, model, ).

- prompting method: have authors consider Chain-of-thought prompting or some other ways to use LLMs for tasks 1&2?

- human performance: expert or turker, how is it collected?

**Relation To Prior Work:**

yes

**Summary And Contributions:**

Authors propose to use Only Conncet's Connecting Wall as a dataset to study machine's "creative problem solving", with "Grouping" (16 -> 4 x 4 words) and "Connections" (explain each group's connection) as two subtasks.

1. In "Grouping", embedding similarity based methods perform badly (best solved <100 out of ~2k groups) , and GPT-3.5/4 performs much better (268/1968 groups solved for GPT-4 1-shot), but still far behind "human performance" (1405/1976).

2. In "Connections", best LLM (10-shot GPT-4) achieve ~40% ROUGE-F1, though still behind ~80% human performances.

---

> ### Author Response · Authors · 2023-08-17
> **Reviewer fDpy response: Revised manuscript with suggested improvements, including better evaluation around fixation effect and clarifications**
>
> We thank you for your cogent summary of contributions and for recognizing our cognitive motivation, and work as an interesting direction and novel for the community.
>
> > better evaluation around fixation effect & Einstellung paradigm:
>
> The opportunity for a better evaluation around cognitive psychology and creativity (i.e., red-herrings, fixation effects) was a common thread among multiple reviewers, and we address this in our *revised manuscript* as outlined in the **global response**. To summarize, we have added additional ablative experiments and results based on two additional datasets (also made available to researchers): **OCW-Randomized** (reduced red herrings), **OCW-WordNet** (zero red herrings), presenting the results in the new *Table 5*. We also extended our discussion in __Section 4.2__ around the result that increasing the number of in-context examples does not always improve performance and speculate why this might be the case.
>
> > prompting method: have authors consider Chain-of-thought prompting or some other ways to use LLMs for tasks 1&2?
>
> *Prompting method*: we were aware of and considered the application of using complex prompting strategies like Trees of Thought or Chain-of-thought prompting (CoT, see __Section 4.2__). CoT-like prompting may improve results, but we concluded that determining the intermediate reasoning steps for the tasks of the OCW dataset is very difficult. To elucidate, it is difficult to enumerate a human's intermediate reasoning steps to solve a wall, and two humans might follow different reasoning steps. We do think this is an exciting future direction for the OCW dataset, and we have expanded the text in __Section 4.2__ to reflect this.
>
> > human performance: expert or turker, how is it collected?
>
> *Human performance*: we regard the reported human performance as **expert**. The results are compiled from observing the performance of the TV show's contestants who knew the format of the tasks and were well-prepared to perform competitively.
>
> We also note and appreciate your *additional feedback* regarding memorization. You are correct that this issue is not a major concern given the poor performance; however, memorization is a very relevant concern from the context of 'data contamination', which is an ongoing research concern (ensuring that LLMs don't simply memorize the dataset in some future iterations). We have updated the __Limitations__ section with a discussion around these points.

---

> > ### Comment · Reviewer_fDpy · 2023-08-27
> > **Thanks**
> >
> > Thanks for the added experiments and clarifications. I've raised my score from 5 to 6.

---

### Author Response · Authors · 2023-08-17
**Global Response to All Reviewers, and Changelog of Revised Manuscript**

We thank all the reviewers for their well-thought-out, relevant feedback on our work and contributions. We especially appreciate the improvement opportunities outlined, as addressing them has greatly enhanced our revised manuscript (uploaded) and overall contributions (added dataset variants). We have coalesced the common threads from multiple reviewers and responded to them in the *Global Response* below.

### Global Response: Additional ablative experiments to analyze the effects of red herrings (fixation effect)

The juxtaposition of language models with human cognitive phenomena was motivational for our work. Thus, we appreciate the reviewers’ suggestion of adding further analyses on the effects of distractors/red herrings.

We first reiterate that every wall of the original dataset (__OCW__) has distractors/red herrings built-in by design. To isolate the effects of red herrings on model performance, we curated two additional datasets with a decreasing number of distractors: **OCW-Randomized** and **OCW-WordNet**. Both are now available via GitHub and HuggingFace repositories as contributions.

In *OCW-Randomized*, we significantly reduce the number of red herrings in each wall by randomly mixing the groups of the original walls. In *OCW-WordNet*, we remove red herrings entirely by creating new groups from the existing ground-truth groups using hyponyms, hypernyms and synonyms from WordNet. The added appendix section (Appendix D) contains pertinent details (including methods and generation).

We added the ablative results on these two datasets in __Section 4.3__ of our revised manuscript. In summary, the results align with expectations and reaffirm our hypothesis: LLMs perform better on the OCW-Randomized dataset, with red-herrings significantly reduced, and even better on toy dataset OCW-WordNet, where the groupings are made obvious using WordNet lexical groupings (synonyms from synsets). This further supports our original hypothesis that LLMs suffer from the *fixation effect*.

### Changelog

The following changes were made to the previous submission version:

1. **Added Appendix D** in the supplementary materials presenting the additional datasets details and ablation experiments results showing red-herrings effects on LLMs and PLMs performances (using the grouping task).
2. **Added subsection 4.3** in the main paper, which contains experimental results on the two new curated datasets, demonstrating increasing performance as red herrings are reduced or removed from the original dataset.
3. Updated __Abstract__ with a summary of [1, 2]
4. __Re-vamped the GitHub/dataset README__ to clearly explain the dataset’s structure and add acknowledgments for the https://ocdb.cc/ fan website and Only Connect quiz show producers and host.

#### Minor changes:
- Minor cosmetic changes (e.g., added \hline to delineate GPT 3.5 and 4 results in pertinent tables).
- Added additional rebuttal references to the bibliography.
- Extended discussion on more complicated prompting strategies (like CoT) in section 4.2.
- Extended the discussion in Limitations of the UK-centric nature of the dataset and the US-centric nature of the LLMs we evaluated.
- Added a discussion in Limitations about the potential of LLMs to memorize walls and the basic steps we took to prevent our dataset from leaking into the training sets of future LLMs.

---

### Decision · Program_Chairs · 2023-09-22

**Decision:**

Accept (Poster)

**Comment:**

This paper creates a new task/dataset for creative problem solving. It builds an evaluation dataset for the task from the quiz show Only Connect. Various baselines evaluation results are reported, including clustering algorithms based on static and contextualized word embeddings and in-context learning based on LLMs, along with human performance results from the quiz show. The results show that SOTA LLMs perform significantly worse than human performance.

Pros:
1) The task to study creative problem solving is very interesting and reasonable, and novel.
2) The evaluations are generally reasonable, clear, comprehensive, and helpful for future research in this direction, providing various baseline results including clustering algorithms and in-context learning of LLMs, along with the human performance baseline.

Cons:
1) The framing of the fixation effect is not sufficiently supported by empirical evidence. Better evaluation around fixation effect & Einstellung paradigm are necessary.
2) The paper should provide analysis of influence of red herrings on the model performance.

In summary, all the four reviewers agreed that this paper is above acceptance threshold and authors also handled concerns from reviewers during discussion period. I recommend acceptance.